

# Swarm Langmuir Probes' data quality and future improvements

Filomena Catapano[1], Stephan Buchert[2], Enkelejda Qamili[1], Thomas Nilsson[2], Jerome Bouffard[3], Christian Siemes[4], Igino Coco[5], Raffaella D'Amicis[6], Lars Tøffner-Clausen[7], Lorenzo Trenchi[1], Poul Erik Holmdahl Olsen[7], and Anja Stromme[3]

[1]Serco c/o ESA, Earth Observation Directorate, Frascati, Italy
[2]Swedish Institute of Space Physics, Uppsala, Sweden
[3]European Space Agency (ESA), Earth Observation Directorate, Frascati, Italy
[4]Delft University of Technology, Delft, The Netherlands
[5]Istituto Nazionale di Geofisica e Vulcanologia (INGV), Roma, Italy
[6]National Institute for Astrophysics, Institute for Space Astrophysics and Planetology, Roma, Italy
[7]DTU Space, Technical University of Denmark, Denmark

**Correspondence:** Filomena Catapano (filomena.catapano@unical.it)

**Abstract.** Swarm is ESA's first Earth observation constellation mission, which was launched in 2013 to study the geomagnetic field and its temporal evolution. Two Langmuir Probes on board of each of the three Swarm satellites provide very accurate measurements of plasma parameters, which contribute to the the study of the ionospheric plasma dynamics. To maintain a high data quality for scientific and operational applications, the Swarm products are continuously monitored and validated via

science-oriented diagnostics. This paper presents an overview of the data quality of the Swarm Langmuir Probes' measurements. The data quality is assessed by analysing short and long data segments, where the latter are selected sufficiently long to consider the impact of the solar activity. Langmuir Probes data have been validated through comparison with numerical models, other satellite missions, and ground observations. Based on the outcomes from quality control and validation activities conduced by ESA, as well as scientific analysis and feedback provided by the user community, the Swarm products are reg-

ularly upgraded. In this paper we discuss the data quality improvements introduced with the latest baseline, and how the data quality is influenced by the solar cycle. The main anomaly affecting the LP measurements is described, as well as possible improvements to be implemented in future baselines.

## 1   Introduction

Swarm is an Earth Observation mission of the European Space Agency (ESA) with the primary objective to measure Earth's

magnetic field and its temporal variations, which enables investigations of, e.g., the core dynamics, geodynamo processes, and core-mantle interactions (Olsen et al., 2013). Further, the Swarm mission is devoted to characterise the ionospheric electric fields, currents, and other ionospheric plasma processes. The space segment consists of three identical satellites, which carry a divers set of instruments to achieve the ambitious mission objectives: a Vector Field Magnetometer (VFM) and an Absolute Scalar Magnetometer (ASM) for collecting high-resolution magnetic field measurements, three star trackers for accurate atti-

tude determination, a dual-frequency GPS receiver for precise orbit determination, an accelerometer to retrieve measurements



of the satellite' s non-gravitational acceleration, and an Electric Field Instrument (EFI) composed of two Langmuir Probes (LPs) and two Thermal Ion Imagers (TIIs) for the plasma and electric field related measurements. The three satellites were launched in 2013 into the same near-polar orbits. Shortly after launch, the satellites were manoeuvred into a constellation in which two satellites, Swarm A and Swarm C, fly side-by-side with 1.4° separation in longitude at the equator at an altitude

of 462 km (initial altitude), and the third satellite, Swarm B, flies at a higher altitude of 511 km (initial altitude). Due to the difference in altitude, the orbital planes precess at different rates such that the angle between Swarm B's orbital plane and that of the other two satellites slowly changes over time. In 2018, Swarm B's orbital plane was perpendicular to that of Swarm A and Swarm C, while by the end of 2021, Swarm B will be counter-rotating with respect to the lower-flying pair, which will result in a conjunction every 47 minutes.

With its plasma instrumentation (LP and TII, (Knudsen et al., 2017)), Swarm is an excellent mission to investigate and survey the ionosphere, its structure and dynamics. Recently, Swarm measurements got more insights on the space weather. As described in Pakhotin et al. (2021), a preference for electromagnetic energy input into the northern hemisphere, on both the day side and the night side, has been reveled by Swarm by averaging the electromagnetic measurements over seasons. As reported by Archer et al. (2019), Swarm EFI measurements also helped to advance the understanding of the auroral phenomenon known

as "Steve", which is visible as subauroral purple emission. Swarm measurements demonstrate that Steve events are not only associated to intense subauroral ion drifts (MacDonald et al., 2018), but also to peaks of plasma temperatures and extremely low densities (Archer et al., 2019). More recently, De Michelis et al. (2021) discussed the possibility to use Swarm data to derive a proxy for the ionospheric turbulence. This work suggests that, looking at the scaling features of the density fluctuations for different locations and geomagnetic activity levels, it is possible to distinguish two families of density fluctuations, one of which

is most probably related to turbulent processes (De Michelis et al., 2021). Furthermore, the long time coverage of the Swarm mission, offers the possibility to perform extended statistical analysis on the climatology of plasma irregularities via plasma-related measurements. The first global statistics obtained by in-situ measurements of plasma variations with Swarm mission, confirmed the presence of three main regions of strong ionospheric irregularities: the magnetic equator extending from post-sunset to early morning, the auroral ovals, and the polar caps (Jin et al., 2020). The long-term behaviour of density gradients

and fluctuations, has been studied by using Swarm data (Jin and Xiong, 2020). This new statistics described phenomena already explored by past missions, but also reveals a new anomaly that is the persistence of strong density fluctuations in the Antarctic during local summer (December solstice) (Jin and Xiong, 2020). The morning overshoot consists in a rapid increase of electron temperature in the early morning hours at low latitudes. Its dependence on geographic regions, local time, seasons and geomagnetic activity has been presented by Yang et al. (2020), by using Swarm data and ISS/FPMU (International

Space Station/Floating Potential Measurement Unit) measurements (Coffey et al., 2008). Plasma density and temperature hemispherical asymmetries have been largely investigated in ionospheric physics, and recently discussed by Hatch et al. (2020). In their study both Swarm and CHAMP (Flury et al., 2006) measurements are used, demonstrating the importance of multi mission synergies and long mission life-time to statistically investigate ionospheric phenomena. For a comprehensive list of scientific results obtained with the support of the Swarm data we remind to the Swarm web-page (ESA, b).





The LPs are relatively simple instruments which are immersed into a plasma to measure electron density, $N_e$, and electron temperature, $T_e$. Owing to their simplicity, relatively small weight and low power consumption, LPs have been used on many satellite missions (Boyd, 1965; Abe and Oyama, 2013). Examples are Demeter (Lebreton et al., 2006), Rosetta (Eriksson et al., 2007), and Swarm (Knudsen et al., 2017). The science data derived from LPs on board Swarm are part of the Level 1B (L1B) products and are obtained from the PLASMA operational processor. The LP data are available at both 2Hz and 1Hz cadence.

The algorithm of the PLASMA processor is described in the L1b Plasma Algorithm document (Buchert and Nilsson, 2018). To support the scientific research the Swarm data products are continuously monitored for Quality Control (QC) and improved by the ESA/ESRIN Swarm Data Innovation and Science Cluster (DISC) Data Quality Team. In this paper the EFI-LP L1B data quality evolution introduced in the current baseline is described and the data quality status is statistically investigated. Known issues and future perspectives are discussed as well.

## 65  2   The Swarm Langmuir Probes

The Swarm LPs have been described by Knudsen et al. (2017) including its "harmonic mode" with the sinusoidally modulated probe bias, in which the instrument is operated most of the times. Also the model equations that are assumed to determine the plasma density, the electron temperature and the spacecraft potential from the currents and admittances computed on board for given biases, are therein included. Complementing the description in Knudsen et al. (2017), we add here a detailed description

of the LP instruments functionalities and operational settings.

The two probes are mounted on the earthward edge of the ram panel as illustrated in Figure 1. They are separated by 30 cm and located relatively close to the faceplate of the TII, which is also mounted on the ram panel. The LPs are expected to provide accurate and independent estimates of the spacecraft potential, which is in principle needed to process the TII data.

The probes are also expected to provide plasma densities and electron temperatures over the entire range of signal magnitudes

encountered along the orbit. The Swarm orbits will cover a limited range of altitudes from about 520 km shortly after launch to 250 km close to re-entry. But they sample practically all latitudes and local times, which results in a relatively large and dynamic signal range. Densities ranging from few hundreds $cm^{-3}$ to several millions $cm^{-3}$ can occur, representing more than 4 orders of magnitude. Avoiding an automatic gain control, which would potentially interfere with reliable and accurate current measurements, both probes are typically operated with fixed but different gains, called low and high gain. The term "gain"

should be understood here rather as a sensitivity of the current measurement than an amplification. The ratio between high and low gain is about 50. High or low gain can be set by a telecommand from ground and usually one of the probes is in high and the other in low gain. Table 1 summarises the differences between the probes and their gain operations.

The surface material of one of the probes is titanium nitride (TiN), which had previously been used for several space missions, for example, Rosetta (Eriksson et al., 2007) and Demeter (Lebreton et al., 2006). Out of concern of the aggressive chemical

reactivity of ionospheric Oxygen ($O$), the other probe surface is made of gold-plated (Au) titanium (Ti) which is a novelty in space. It is known that Ti is very difficult to electroplate (e.g., ENS-Technology), however, a small company with experience in gold-plating jewellery made of titanium was given the contract. Testing before launch did not reveal any problems with the





| Probe | Gain | Surface | Position |
|-------|------|---------|----------|
| 1 | high up to Dec 2019- low onward | TiN | $-\hat{y}$ |
| 2 | low up to Dec 2019- high onward | Au | $+\hat{y}$ |

**Table 1.** List of differences between the two LPs on each Swarm satellite. The probe position is defined with respect to the spacecraft coordinate system where $\hat{x}$ is along the fly direction, $\hat{y}$ horizontally crosses the satellite toward local dusk, and $\hat{z}$ points toward the Earth.

Au probes even after baking with temperatures of up to 300° C and after exposure to ultra-sound. Presently, it is unknown how much Au is left on the Ti surface after more than 7 years in space.

## 3   LP data processing

The L1B PLASMA processor, which is used to generate the LP data products, is organised according to simple flow-chart reported in Figure 2. It uses as inputs the L1B products containing position and velocity of the satellite, auxiliary data, and EFI-LP Level 0 (L0) data, to obtain three L1B and one Level 1A (L1A) data products. The auxiliary data contains information that support the Swarm data processing, such as geomagnetic indices, or instrumental calibration parameters obtained during ground tests. The L0 data contains raw measurements from each Swarm instrument and are essentials to generate the L1 products. The EFI-LP L1A product (EFIX_LP_1A) contains information about the LP configuration, ion and electron currents in different regimes, and bias voltages. The L1B product LP_X_CA_1B delivers the LP calibration parameters derived for each probe by the L1B PLASMA processor. Finally the EFIX_LP_1B and EFIXLPI_1B products, provide the plasma parameters as density, electron temperature, plasma potential, together with the spacecraft position and the flags indicating a possible source of error for each data point. The EFIX_LP_1B are available at 2 Hz sample rate. By interpolating these products at exact UTC, the EFIXLPI_1B products are obtained at 1Hz sample rate. Also, the LPs operate in different modes. The "harmonic mode" (HM) consists of sinusoidal varying biases applied to the LPs. Each HM cycle lasts for 0.5 s, and during the HM currents and admittances are measured. To our knowledge this method to obtain the current-voltage (I-V) characteristic of the space plasma is being used in orbit for the first time. The HM operates most of the time, while the classical "sweep mode" occurs each 128 s, and lasts for 1 s. In sweep mode the I-V curve is measured traditionally by scanning the probe bias over a range that expectantly stretches from a dominant ion current (at negative bias) to a saturated electron current (positive bias). Sweep mode data are not used in the L1B PLASMA processor, but are separately analysed and provided as an additional "advanced" product. Furthermore, each six hours a calibration mode is activated and a calibration data packet is generated on-board. This data are used for calibration purposes and, during the calibration mode, short data gaps are registered in L1B PLASMA products. The EFI-LP data products, and the other Swarm L1B products, are provided in daily files with a latency of four days. Detailed information on Swarm L1B processors and data products are described by DTU (2019a, b). The Swarm products are freely accessible through the ESA dissemination server (ESA, a). In the next two sessions are described the recent data products evolution and data quality characterization.





### 3.1 Evolution from product baseline 04 to 05

The product baseline is a number identifying the data that were generated in a consistent way, i.e. using the same algorithms and input parameters, and, thus, constitute a data set. The first two of the last four digits of the product filename indicate the baseline. The product baseline is incremented when algorithm or input parameter upgrades lead to significant improvements in the data quality of the related products. The first PLASMA baseline went into operation in 2015 with the number 04. Before baseline 04, LP data were processed with a provisional processor by the Swedish Institute of Space Physics (IRF) (ESA, 2015).

When the final version of the PLASMA processor was ready to be transferred into operation, it was deployed directly with baseline number 04 to be aligned with the other Swarm processor baselines. Thus, baseline lower than 04 are not available for EFI data products. Since September 2018, the PLASMA baseline has the number 05. An updated version of this processor has been deployed in operation in February 2020 containing only minor evolution, thus the baseline number remained unchanged. A complete description of all the evolution introduced with these processors is reported in the related technical notes ESA

(2018, 2020b). In the following we will discuss the major differences in PLASMA products between the baselines 04 and 05, consisting on the electron temperature ($T_e$) computation from high-gain probe, and the decoupling of PLASMA processor from MAGNET processor.

### 3.1.1 Electron temperature computation from the high-gain probe

Each of the two LPs on board the Swarm satellites can be commanded to high or low gain. By electronically coupling a second

shunt resistor in parallel the mode is low gain which allows higher probe currents to be measured without ADC (Analog Digital Converter) overflows. Typically, one probe is set to the low gain and the other one to the high gain. The LP product parameters can be estimated from each probe. In practice the values often differ which we suspect is because of the different probe gains.

The first analysis, preceding baseline 04, estimated the electron density $N_e$ and electron temperature $T_e$ from the high gain probe for low densities/probe currents, from the low gain probes for high densities/probe currents, and by blending the results

from both probes for an intermediate range of density/probe current. This avoided sudden jumps which would be caused by switching the probes at threshold values. Typically the low gain probe needs to be used at the dayside magnetic equator because of very high density in the ionization anomaly, and the high gain probe is more appropriated for other regions. In the commissioning phase it became clear that the regularly occurring transition between probes produced unphysical variations of the estimated parameters even when smoothed by the intermediate blending. Therefore the algorithm to estimate the den-

sity was changed to use the weaker ion current instead of the retarded and saturated electron currents. The ion current and admittance is always and very reliably measured by the high gain probe. The density product is therefore rather an ion density product, though often designated still $N_e$. At Swarm altitudes, in the thermosphere and F region, the ion and electron densities are expected to be equal (only in the mesosphere and D region negatively charged ions and dust particles could cause $N_e$ to be less than the positive ion density). Also for $T_e$ the blending of high and low gain estimates was eventually abandoned in

order to avoid producing unphysical variations at transitions. This, however, has the drawback, that especially in the ionization anomaly ADC overflow occurs in the high gain saturated electron current. The $T_e$ from the low gain probe is dropped in, with





| $< \Delta Te/Te_{05} >$ | | | |
|---|---|---|---|
| Swarm A | Swarm B | Swarm C | MLT |
| - 11.6 % | 2.31 % | - 5.52 % | ALL |
| - 9.56 % | 1.01 % | - 5.04 % | [8-16] hr |
| - 14.1 % | 2.94 % | - 6.24 % | [20-4] hr |

**Table 2.** Average relative difference between $Te_{04}$ and $Te_{05}$ for all the Swarm spacecraft for different MLT ranges. The results are obtained considering one week of data from 7 to 13 September 2018.

a flag value as warning. This modification has been introduced with baseline 05. The region characterized by larger plasma density are generally observed at equatorial and low latitudes. In particular, in correspondence to day side equatorial crossings, it is possible to observe the typical double peak of the plasma density. This feature is related to the equatorial fountain effect characterizing the equatorial ionisation anomaly (Kelley, 2009). Also, the ADC overflows are frequently observed at equatorial latitudes. Thus, to compare the measurements from baseline 04 (where high and low gain $T_e$ measurements were blended together) and baseline 05 (where only high gain measurements are used) it is worth to consider the latitudinal variation. Figure 3 shows the differences between $T_e$ obtained from baseline 04 ($Te_{04}$) and baseline 05 ($Te_{05}$) as a function of Quasi-Dipole (QD) latitude. The analysis is shown separately for (a) day side, (b) night side, and (c) full Swarm A orbits during one week in September 2018. The different phases of the orbits have been selected with respect to the magnetic local time (MLT). The analysis is limited to the latitudinal range to +/- 50° because at higher latitudes the electron temperature has a level of fluctuations too strong to obtain a meaningful comparison between $Te_{04}$ and $Te_{05}$. Figure 3 demonstrates that $Te_{04}$ is on average larger that $Te_{05}$ at higher latitudes. On the day side, the two baselines are comparable at equatorial latitudes (panel (a)), while the differences in this region are larger on the night side (panel (b)). In particular, the night side presents a negative peak between $-10$ and 10 degrees of QD latitude. Also, in Figure 3 (c),we note a negative peak in correspondence to equatorial latitudes, and a decrease for higher latitudes. Table 2 reports the average relative differences $< \Delta Te/Te_{05} >$ for each MLT range, where $\Delta Te = Te_{04} - Te_{05}$. The results demonstrate that, on average, the baseline 05 measures $T_e$, which is 5-10% larger for the lower pair (Swarm A and C). This is a very good improvement, because it has been shown that the LP measurements of baseline 04, on average, underestimate the electron temperature with respect to ground measurements (Lomidze et al., 2018). Thus, the larger $T_e$ measurements obtained with baseline 05 represent a better agreement with ground observations.

### 3.1.2 Decoupling between PLASMA and MAGNET processors

In the previous configuration related to baseline 04, the PLASMA processor had a dependence on the MAGNET and OR-BATT processors. The ORBATT processor is fundamental for the L1B processing chain because it generates the L1B satellite ephemeris and attitude products, which are inputs for all the other processors. The MAGNET processor generates L1B magnetic measurements data products, which also contain the satellite position and attitude for convenience. The PLASMA processor needs as inputs the spacecraft position and velocity expressed in the Earth-fixed reference frame. In baseline 04, the spacecraft





velocity was retrieved from the ORBATT processor, while the spacecraft position was retrieved from the MAGNET proces-
sor. Also, in baseline 04, magnetic measurements from MAGNET processor were needed to compute electrical field from TII
measurements. This dependence on other processors implies that if one of those has a partial or total failure in producing the

data products, then also the PLASMA processor fails. However, it was observed that the dependence on the MAGNET proces-
sor was not necessary, since the generation of the electrical field from TII measurements where removed from the PLASMA
processor in baseline 05. Therefore, in the latest baseline, satellite position and attitude data can be directly retrieved from the
ORBATT data products. As a consequence, with baseline 05 PLASMA processor is decoupled from MAGNET, and it is now
depending only on the ORBATT processor. This decoupling offered the opportunity to recover past data gaps occurred because

of MAGNET failures. In particular, with baseline 05 it was possible to recover the production of 4 days for Swarm A, 11 days
for Swarm B, and 5 days for Swarm C. A full list of recovered data products is available in ESA (2020a). Even if it is a very
small portion of data that has been recovered over more than seven years of Swarm measurements, this still represents an im-
provement introduced with respect to the older baseline 04. Finally, we note that the decoupling of PLASMA from MAGNET
processor has no impact in the LP data quality.

**3.2  Baseline 05**

The baseline 05 covers the data products from December 2013 until today. The LPs on board Swarm, can well capture the
ionospheric variability in shorter intervals of time. Figure 4 shows the variation of plasma density and electron temperature as
measured by Swarm B. Invalid measurements are removed in this Figure. The missing data at equatorial latitudes in Figure 4
(b) are mainly due to ADC overflows, which generate invalid measurements. An interesting feature, that is worth to be noticed,

is the typical double peak of the electron density at equatorial latitudes. This effect is related to the equatorial electrojet fountain
(Kelley, 2009), and it is well visible in the Swarm measurements. In fact, at mid-low latitudes the density is higher, showing
two peaks at around ±10 degrees of QD latitude, and slightly lower values at around zero degrees. At higher latitudes, the
density is lower again. The electron temperature instead, presents a different features showing lower values at mid- and low-
latitudes, and higher values at higher latitudes. This is another typical characteristic of ionospheric plasma (Kelley, 2009).

During more than seven years in orbit, the Swarm measurements span more than half of a solar cycle. Figure 5 reports the sun
spot number and the F10.7 index as indicators of the solar activity, and the Kp index as an indicator of geomagnetic activity.
The regions highlighted in orange represent the years when Swarm is in-orbit. Such a long temporal coverage with Swarm
measurements opens the opportunity to study the impact of solar activity on the ionosphere (Xiong et al., 2010) and to perform
a long-term analysis of the ionospheric variations as well as multi-mission studies (Noja et al., 2013; Xiong et al., 2020). Here

we discuss the data quality variation of Swarm LP measurements with respect to the last solar cycle. The F10.7 index is used
as reference for the solar activity (Covington, 1947, 1948; Tapping, 2013). The F10.7 index is a proxy for the solar EUV flux,
which is the dominating source of ionisation, molecular dissociation, and heat in the thermosphere-ionosphere (see for example
Liu and Chen (2009); Vaishnav et al. (2019)). Figure 6 shows the average plasma density variation from December 2013 to
July 2020 as measured by Swarm A, separately for the ascending and descending orbit phases in panel (a) and (b), respectively.

Panel (c) shows the F10.7 index for the same interval of time. The density profile shows a high correlation with the F10.7



index. The solar radiation is the fundamental driver of density and temperature variations in the ionosphere (Prölss, 2004; Kelley, 2009). An example are the different characteristics of ionospheric plasma on the day and night side, the latter having lower densities and higher temperatures (see for example Heelis and Maute (2020) and reference therein). Thus, the strong correlation between F10.7 index and Swarm density measurements reported in Figure 6 , which is related to the ionospheric
processes driven by the solar activity, represents additional evidence of the good quality of the Swarm data.

Each LP data point is associated with a flag indicating the instrument performance and settings, together with the source of possible errors. For more information on the flag we refer to Section 6.8 of the document DTU (2019b). The percentage of measurements that are flagged as invalid is a useful proxy for the data quality and instrument performance. A larger percentage of invalid measurements obviously indicates a poorer data quality. Figure 7 reports the daily and monthly percentage of invalid
measurements from the beginning of the mission up to July 2020, for Swarm C. The results are reported for the plasma density and electron temperature in panel (a) and (b), respectively. The shadowed area in the panels represents the F10.7 index variation in the same period. In Figure 7 (b), we observe a common trend in the percentage of invalid measurements of electron temperature $T_e$ and the F10.7 index, whereas the opposite trend is visible for the percentage of invalid measurements of plasma density in Figure 7 (a). These trends are similarly observed for all three Swarm satellites. During the solar minimum, the
plasma density decreases, as also shown in Figure 6. In particular, the LP measures negative plasma densities more frequently during the solar minimum. This feature is reflected in Figure 7 (a) by a larger number of invalid measurements at lower F10.7 index values. During periods of stronger solar activity we observe more frequently ADC overflows, which generate invalid $T_e$ measurements. This feature is represented in Figure 7 (b), by a larger number of invalid $T_e$ measurements at lower F10.7 index. The geographical location and temporal variation in the occurrence of the invalid measurements, are very useful to the Swarm
EFI-LP team to study the instrument performance and to detect possible anomalies in the LP measurements.

The plasma density, can also be derived from the faceplate (FP) on-board Swarm as part of the TII instrument. The FP, similarly to a planar Langmuir Probe (Oyama, 2015), measures the current with a cadence of 16 Hz. The electron density $N_e$ is derived from FP current measurements only for certain orbits per day, namely when the TII is not active. The FP data and relative technical notes are available for all Swarm users (see IRF (2017)). A validation of LP density measurements can be
performed by comparing the LP and FP derived densities. Figure 8 shows a scatter plot between density as measured by the LP and FP separately for the (a) day and (b) night. We observe a very high correlation of 0.98 between the two data sets for the day side and a moderate correlation of 0.47 for the night side. The relative difference between the FP and LP density measurements, defined as $(Ne_{FP} - Ne_{LP})/Ne_{LP}$, is 19% for the day side, and 34% for the night side, noting that the FP density measurements are generally higher than the LP density measurements. In this context, it is worthwhile to emphasise that the LP processor
algorithm makes the assumption that ionospheric plasma at Swarm height only contains $O^+$ ions, while the FP data processing does not need assumptions on the plasma composition. Thus, the discrepancy between the FP and LP measurements could be caused by the different assumptions on the plasma composition, in particular for the night side. Indeed, the electron density in nocturnal regions is lower compared to the day side, which is due to the weaker sun illumination and consequently lower ionisation on the night side, and larger number of molecular ions (Kelley, 2009; Heelis and Maute, 2020). The comparison
between the FP and LP density measurements demonstrate that the two data sets are in good agreement. The results also show





that the L1b PLASMA algorithm can be further improved by taking the difference in the ion composition between the day and night side into account. Possible ways to improve the plasma density computation are under investigation and will be included in future baselines.

## 4 Known issues and future plans

The Swarm LP measurements are of very good data quality and have a high value for scientific investigations. However, few anomalies affect the LP measurements which are continuously monitored and investigated by the ESA Data Quality Team and the scientific community. The source of these anomalies is only partially understood, which leaves open questions in both physical and instrumental domains. This section is dedicated to the description of the occurrence of one of these anomalies, namely the occurrence of extremely high values in the electron temperature measurements, which has a large impact on the on

data quality and scientific investigations. In addition, we describe the calibration of LP measurements, which will be introduced in the next baseline 06.

### 4.1 Extreme $T_e$ values

The ionospheric electron temperature typically ranges from a few hundred Kelvin during quiet periods at lower latitudes to a few thousands Kelvin during extreme events such as Steve auroral emissions (Archer et al., 2019), during which peaks of 8000

K were observed. However, the LP on board Swarm satellites occasionally measures $T_e$ values up to more than twenty thousand Kelvin, which have to be considered as "extreme". Figure 9 reports the extreme $T_e$ values that occurred in 2019, as a function of the solar elevation ($\alpha$) and azimuth ($\beta$) angles with respect to the spacecraft. The extreme $T_e$ values represent around 0.1% of the data in 2019. In particular, about 19% of the extreme $T_e$ values are located between $\pm50°$ of QD latitude (green circles denoted EQ in the legend), 15% are located at latitudes higher than $50°$ (purple circles, NH in the legend), and 65% are observed

at latitudes below $-50°$ (blue circles, SH in the legend). It is worthwhile to notice that the distribution is more scattered for positive $\alpha$ values, i.e. when the sun illuminates the spacecraft from the rear (anti-flight direction). On the opposite, we observe a more ordered distribution for negative $\alpha$ values, i.e. when the sun shines on the front of the satellite. Similar results are obtained for all three Swarm satellites (not shown). This peculiar behaviour suggests that part of the $T_e$ extreme values are probably related to instrumental disturbances possibly triggered by the sun illumination. Table 3 reports some statistics on the

occurrence of extreme $T_e$ values at different latitudes, which were observed in 2019. Numerous investigations are ongoing in order to identify the source of these extreme $T_e$ values, which are more frequently observed in the southern hemisphere, as reported in Table 3, and occur at specific angles of the solar illumination of the spacecraft. The extreme $T_e$ values are currently flagged as valid measurements. In the next baseline will be introduced a dedicated flag value to highlight this anomaly.

### 4.2 LP calibration against ground measurements

The Swarm LP data have been extensively compared with other data sets during past years. For example, LP data have been compared with Digisonde (Singh et al., 2021), other missions (Liu et al., 2020), and with the International Reference Ionosphere





| % of $T_e$ extreme values | | | |
|---|---|---|---|
| Swarm A | Swarm B | Swarm C | QD-Lat |
| 0.09 | 0.07 | 0.15 | ALL |
| 19.2 | 10.1 | 4.9 | [-50 50]° |
| 15.4 | 18.4 | 16.4 | > 50° |
| 65.3 | 71.6 | 78.6 | < -50° |

**Table 3.** Percentage of $T_e$ extreme values ($T_e > 20000$ K) observed during the **2019** at different latitudinal locations.

model (IRI, Bilitza (2018)) during quiet as well as disturbed periods (Pignalberi et al., 2016). Swarm LP measurements also contributed to ionospheric modelling, as described in Pezzopane and Pignalberi (2019). Furthermore, Swarm measurements have been statistically validated as presented in Lomidze et al. (2018), by comparing LP data from December 2013 to June
2016 with nearly coincident measurements from low- and mid-latitude incoherent scatter radars (ISRs). The ISR measures altitude profiles of ionospheric plasma and temperature. The ISR measurements usually extend beyond the altitude of the Swarm satellites, thus making them well suited for validation studies. The results demonstrate that Swarm LP measurements underestimate the plasma density by approximately 20% and overestimate the electron temperature by approximately 400 K. The results of Lomidze et al. (2018) allow to calibrate the Swarm LP density and electron temperature measurements. The
calibration parameters represent the correction to LP data to obtain a better agreement with ground measurements. As discussed in section 3.2, the comparison between LP and FP data revealed a difference of the 18% in the plasma measurements on day side. At the present stage of the development, the calibrating the LP measurements yield a much better agreement between LP and FP density, where the remaining difference is only $\sim 3\%$. The calibrated LP measurements will be very useful for future studies dedicated to the comparison of the Swarm LP data with other data sets. The calibration of the LP measurements will be
implemented in the future baseline 06, where the difference between measured and calibrated LP data will be stored in a new variable in the L1B EFI-LP data products.

## 5   Conclusions

The quality control and validation activities performed by the Data Quality Team in the frame of the ESA Swarm DISC reveal the good quality and instrument performance of the Langmuir Probes on board the Swarm satellites. The analysis
demonstrated that the current baseline 05 plasma data products are substantially improved with respect to previous baseline 04. In particular, the electron temperature measurements are more stable and, on average, smaller with respect to the older baseline. The changes introduced in the current baseline lead to the recovery of past data gaps, increasing the data coverage and reducing the possibility of future failures for generating the data. The LP measurements have captured the ionospheric plasma variability over more than half of a solar cycle, which revealed that the data quality depends on the solar activity. In particular, plasma
density measurements are more accurate during higher solar activity. On the opposite, electron temperature measurements are





more stable during low solar activity. These results are highly related to the LP instrumental settings and are well tracked by the monitoring of the data quality. Plasma density LP data have good agreement with TII faceplate measurements, particularly on the day side. However, the comparison between the two data sets demonstrates a weaker correlation on the night side. The disagreement in nocturnal regions is partially related to the fact that the LP processing algorithm assumes that the plasma is

composed of singly ionised oxygen only. Investigations are ongoing in order to include molecular ions in the plasma algorithm and to improve the quality of the plasma density computation in future baselines. The next release of the L1B LP data products will include the calibration parameters for plasma density and electron temperature, which are statistically derived from a comparison with ground measurements. Furthermore, a dedicated flag will be introduced to identify the extreme values of electron temperature. These changes will further improve the quality of the Swarm LP L1B data products and will further

promote their application to a broad range of ionospheric studies.

## 6 Data availability

In accordance with ESA Earth Observation Data Policy, all Swarm Level 1b and Level 2 products are freely accessible to all users at the site swarm-diss.eo.esa.int accessible via https://.

*Author contributions.* This study was led and coordinated by FC and SB with contributions and internal review by all named authors.

*Competing interests.* The authors declare that no competing interests are present.

*Acknowledgements.* This study has been supported by the Swarm DISC project funded by ESA through contract No. 4000109587/13/I-NB.





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





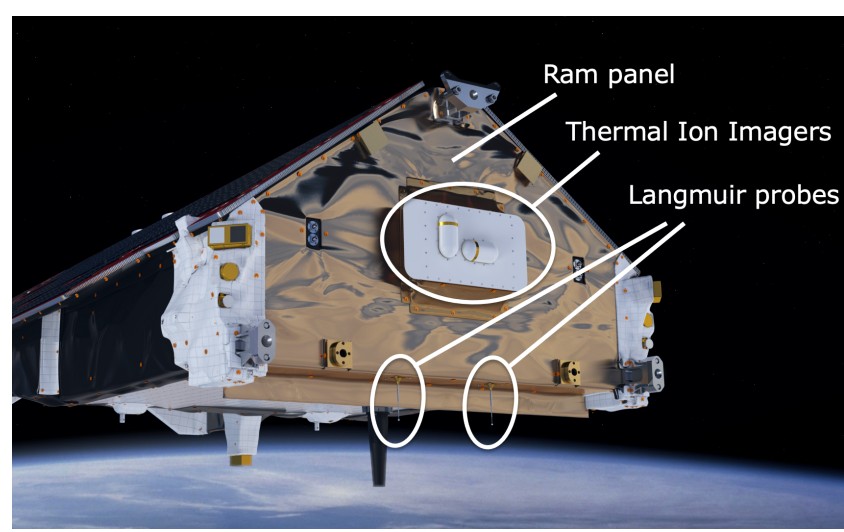

**Figure 1.** Location of the LPs below the ram panel. Image credits: ESA/ATG medialab.





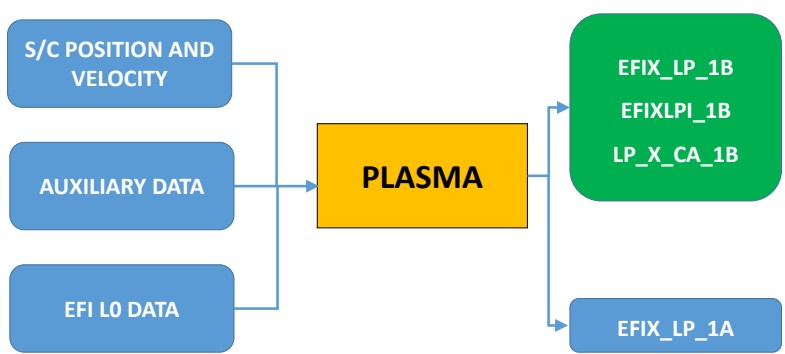

**Figure 2.** Schema of the L1B PLASMA operational processor. The blue boxes on the left side represent the input files, the central yellow box represents the PLASMA processor, and the right side boxes represent the processor outputs. In particular, the green box contains the EFI-LP L1B data products.



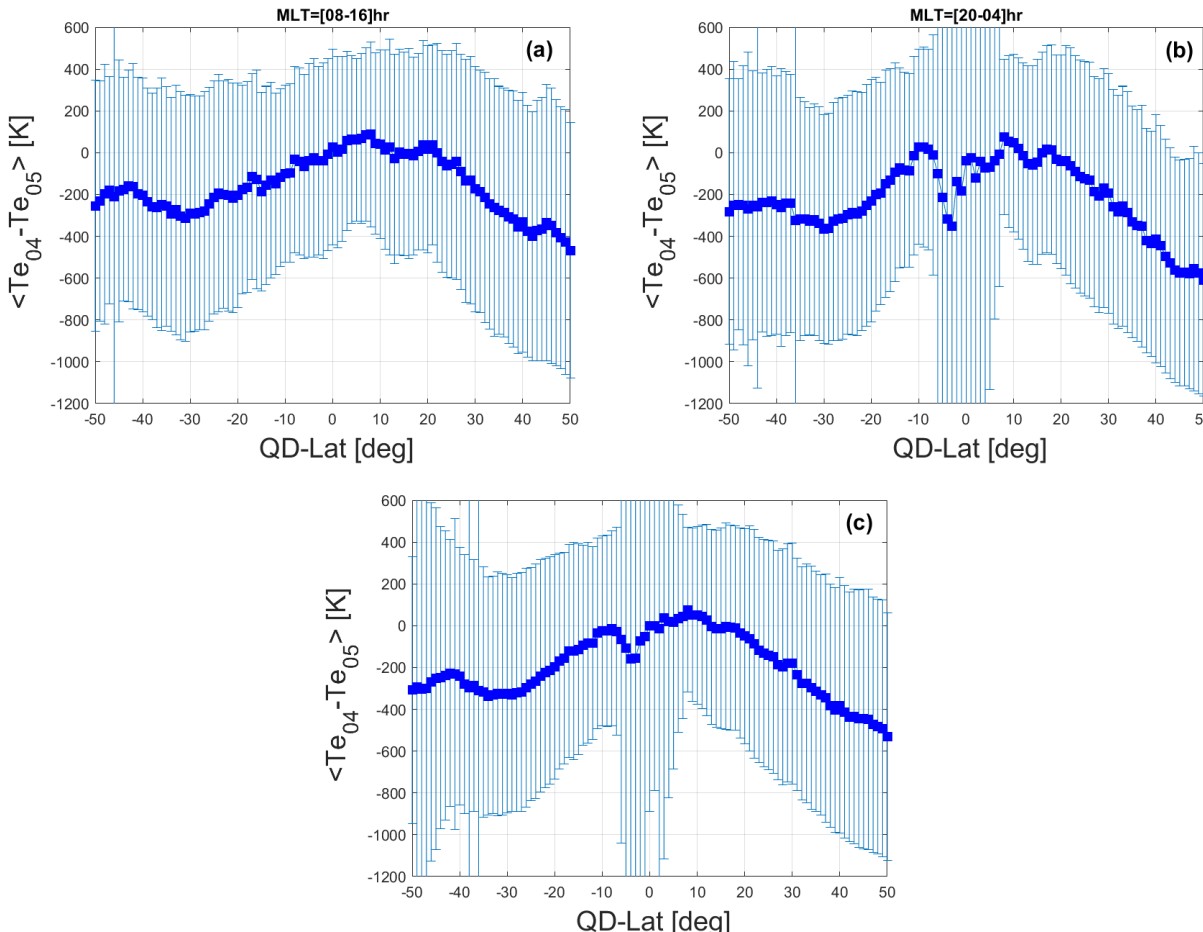

**Figure 3.** Difference between electron temperature $T_e$ computed from baseline 04 ($Te_{04}$) and 05 ($Te_{05}$) as a function of quasi-dipole latitude, measured by Swarm A from 07-09-2018 to 13-09-2018. Blue squares are the daily averages of each one-degree bins in latitude, while vertical bars represent the standard deviation. The figure shows the difference during (a) day side, (b) night side, and (c) full orbits.



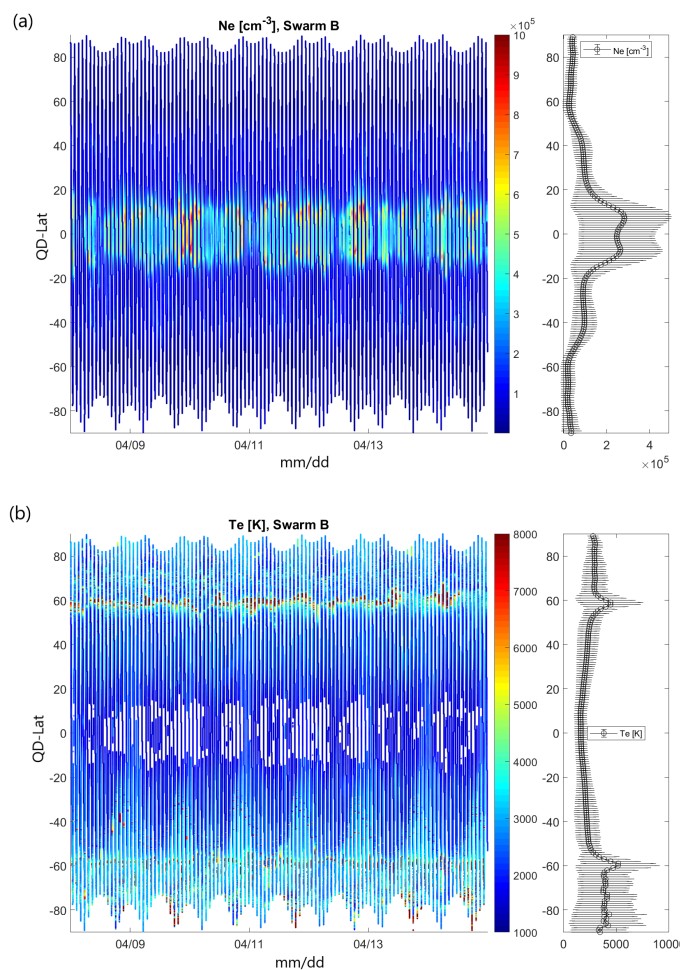

**Figure 4.** Plasma density (a) and electron temperature (b) measured on board Swarm B between 08/03/2018 and 15/03/2018, as a function of Latitude in quasi dipole coordinate, and time. The vertical lateral panel shows the average (squares) and standard deviation (vertical bars) for each degree in latitude. During this period the spacecraft was performing a noon-midnight orbit.

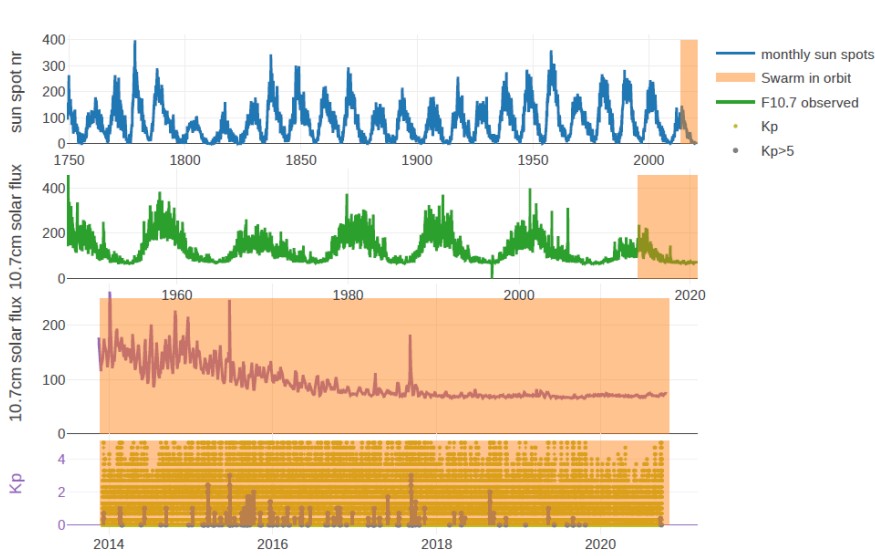

**Figure 5.** Overview of common indices for solar activity intensity. From top to bottom: sun spot number, F10.7 index, and Kp index. The region highlighted in orange represents the period with Swarm in orbit.



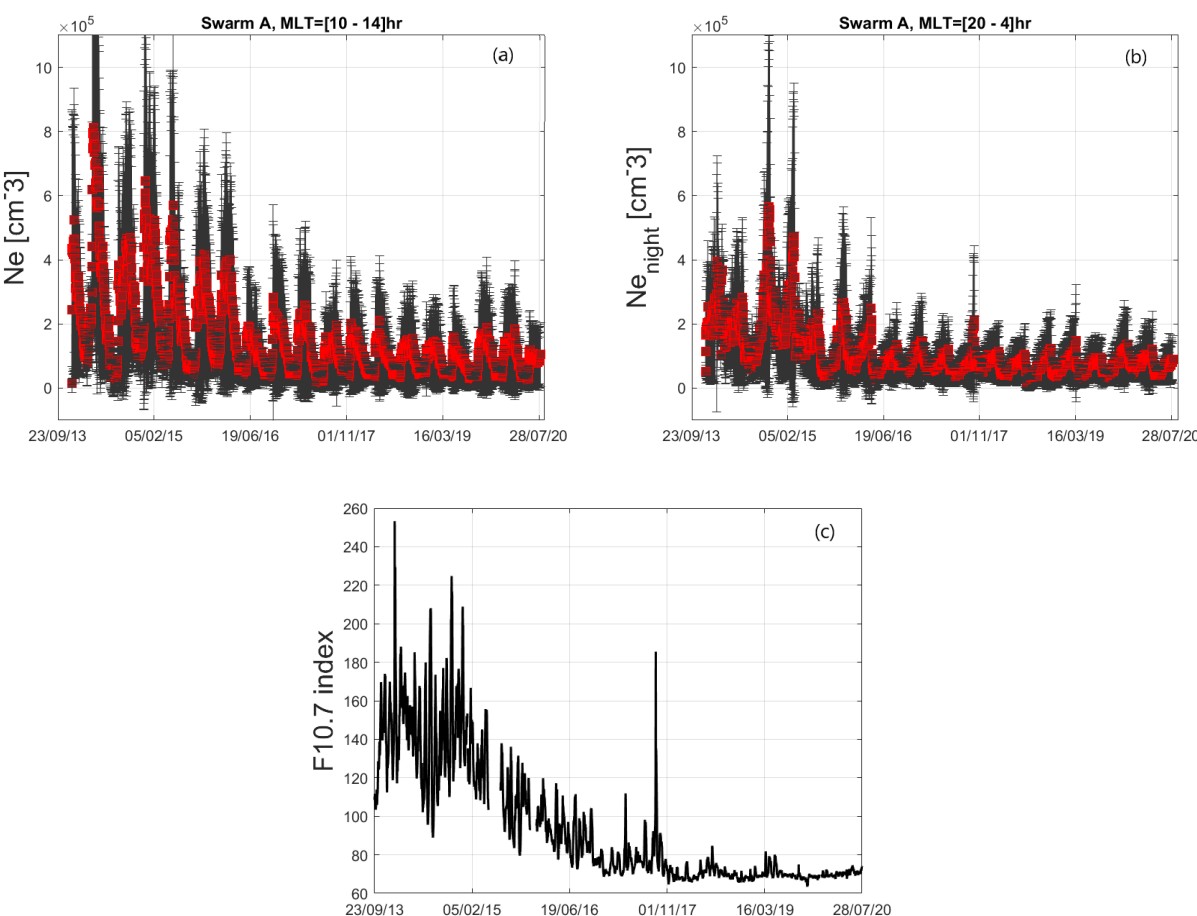

**Figure 6.** Orbital averages of the electron density variation (red squares) with standard deviations (vertical bars) during (a) ascending and (b) descending orbit phases, observed by Swarm A from December 2013 to July 2020. Panel (c) displays the F10.7 index in solar flux units for the same period. Similar results are obtained also for Swarm B and C (not shown).



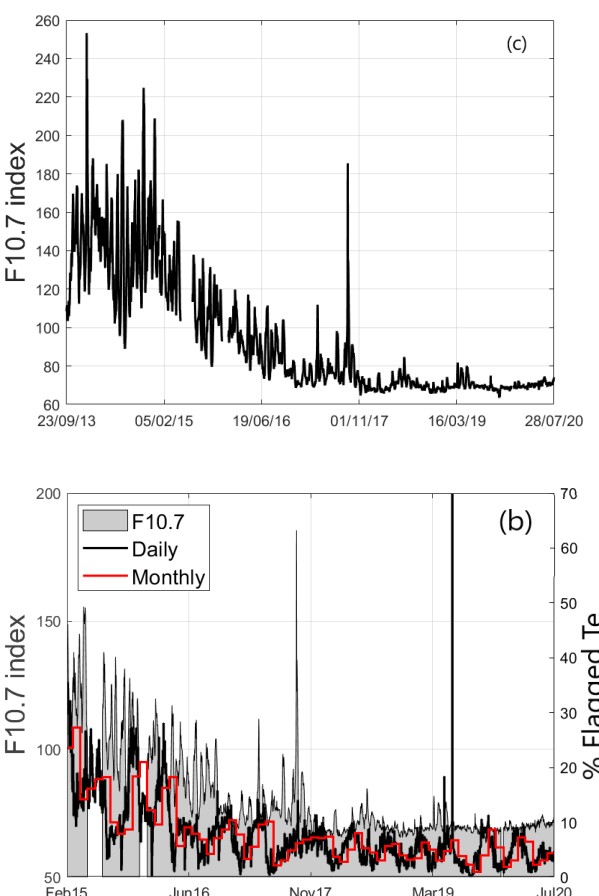

**Figure 7.** Daily (black lines) and monthly (red lines) percentage of invalid measurements (right axis) of (a) plasma density and (b) electron temperature measured by Swarm C from February 2015 to July 2020. The grey area in the panels represents the F10.7 index (left axis) in solar flux units in the same interval of time.





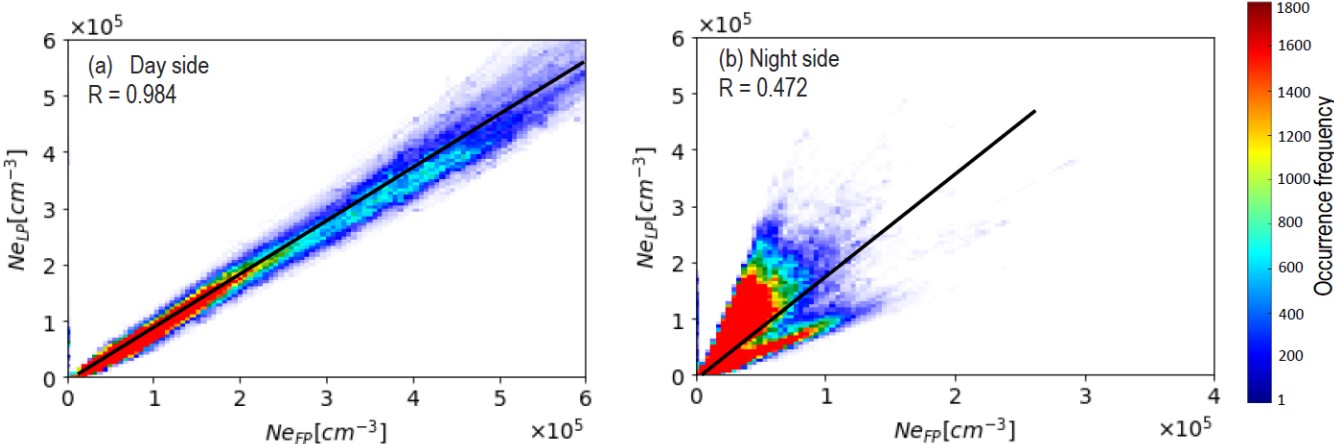

**Figure 8.** Comparison of the plasma density derived from the LP and FP on the (a) day side and the (b) night side as measured by Swarm C in February 2020. The black lines represent the linear fit obtained for the two data sets.

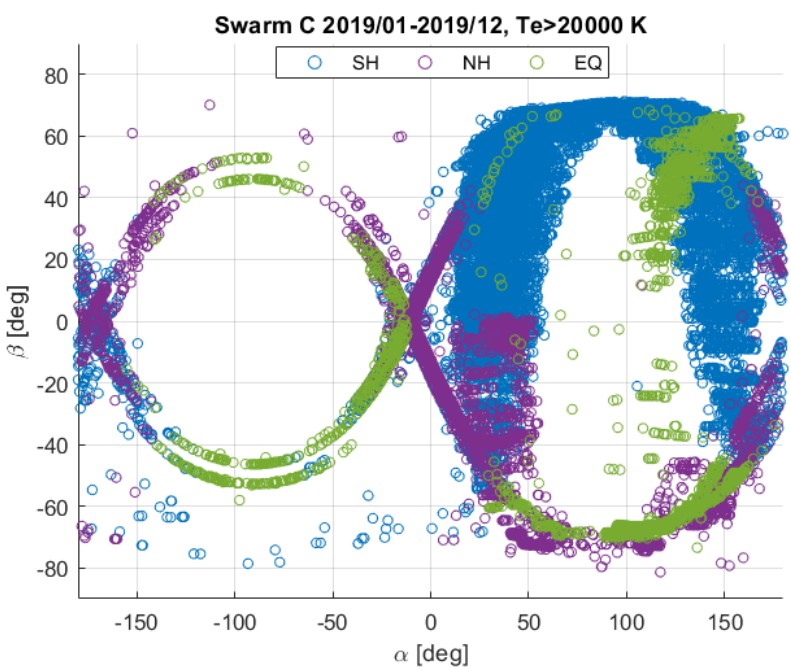

**Figure 9.** Electron temperature ($T_e$) extreme values as a function of solar elevation ($\alpha$) and azimuth ($\beta$) angles as observed by Swarm C in 2019. Measurements located at latitudes between $\pm 50°$ in QD coordinates are represented with green circles (EQ). Measurements at latitudes smaller than $-50°$ are represented by using blue (SH) circles. Purple circles (NH) denote measurements at latitudes larger than $50°$.