# Peer review of "Swarm Langmuir Probes' data quality validation and future improvements"

_Geoscientific Instrumentation, Methods and Data Systems, 2021_

## Referee Comment (RC2)

**Swarm Langmuir Probes' data quality and future improvements**

Filomena Catapano , Stephan Buchert , Enkelejda Qamili , Thomas Nilsson , Jerome Bouffard , Christian Siemes , Igino Coco , Raffaella D'Amicis , Lars Tøffner-Clausen , Lorenzo Trenchi , Poul Erik Holmdahl Olsen , and Anja Stromme

Reviewer's detailed comments

§
Line 2:
« very accurate » needs to be quantified. One knows that LP measurements are not that accurate in absolute terms.

Line 4: Clarify the « operational use » that is being made of LP measurements
Line 9: Replace « conduced » by « conducted »

Line 9: Note sure that how the feed back from the user community is addressed in the manuscript. It should be noted that the intend to involve the feedback from the science community is excellent and will certainly contribute to the data product quality.

Line 12: add « in the derived plasma parameters » after « improvements »

Line 17: I am wondering whether the accurate determination of the plasma parameters contributes to the improvement of the magnetic field measurement analysis? If not, please ignore my question, if yes it would be interesting to say a few words on the subject.

Line 18: Replace « divers » by « diverse »

Line 31: Rephrase « insights on the space weather »

Line 33: Replace « reveled » by « revealed »

Line 41: remove the comma after « mission »

Line 41: I doubt the word « climatology » is the right word. Would « meteorology » be more appropriate? as it seems related to space weather effects.

Line 42: add the word « performed » after « variations »

Line 45: replace « this new statistics » by « This new statistical study »

Line 47: Precise « Antarctic » which usually refers to the continent

Line 59: Provide a reference to the « PLASMA operational processor »

Line 60: Is « PLASMA processor » the same thing as « PLASMA operational processor »?

Line 61: add a comma after « research »

Line 63: Precise if you refer to « data quality » or « Plasma parameters quality »

Line 68: Explain « admittance »

Line 69: Precise « given biases »

Line 71-73: The position of the LP sensor real close to the Satellite skin probably affects the accuracy of the spacecraft potential measurements (and probably to a certain extent plasma density and temperature) , as the LPs are most likely inside the plasma sheath of the satellite. The Debye Length of the effect should be discussed. This is a potential issue which is not addressed in the paper. Perhaps it was addressed in a reference publication. Its impact on the accuracy of the plasma parameters measurements should be addressed in this paper.

Line 74 (and 77): Replace « signal » by « current »?

Line 78: replace « interfere » by « interfer »

Line 82: What was the rationale for the gain switch between the two probes in 2019?

Line 84-86: The sentence « Out of concern »… ; please discuss whether if there is a difference in terms of degradation between Au and Ti due to Atomic oxygen.

Knowing the subject, I may add that, one of the concerns, was also the oxidation of TiN on ground.

Line 87: Explain what « testing before launch » was made

Line 90 and following

It's rather hard for the reviewer (and will most likely be hard for the readers) to follow the meaning of all acronyms used. A list of acronyms would be useful.

Line 103-104. I am myself not aware of a previous use of the HM method in space. I am wondering how the set point of the applied bias signal (and its amplitude) is selected and adjusted as satellite potential must significantly vary along the orbit. I note that it is written that « Sweep mode data are not used in the PLASMA processor… ». Has a comparison of the plasma parameters regularly obtainable from the I-V sweeps and the HM method been made.?

Line 112: Replace « sessions » by « sections »

Line 115 and following

Would it be useful to summarize in a table, the differences between the baselines 04 and 05?

Line 129-130. Clarify « second shunt resistor ». What's the role of the first one?

Line 132: The sentence « In practice the values often differ which we suspect is because of the different probe gain » is problematic. The sudden jumps mentioned in line 135 should be further discussed. Also, statement in line 138-139 casts doubts on the accuracy of the measurements at the gain transition.

It seems to point to the fact that the measurements performance in high and low gain are not fully understood, thus question the validation of the derived plasma parameters. It may be useful to look at this issue by looking at the regular sweeping I-V characteristics.

Line 139 and following: Provide details on the method used to derive the ion density rather than the electron density. This should be supported by modelling the ion sheath in the vicinity of the LPs. Assumption the Ni=Ne at the LP location may be questioned.

Line 147-148: « .. the region (singular) . ; are (plural) «  . Correction needed.

Line 147: « Larger » than what? use « large «?

Line 176: Replace « where » by « were »

Line 178; add «, the » after « 05 »

Line 179: Add « the » after « gaps »

Line 186: replace « today » by a specific date

Line 210: How « good » is good?. Quantify goodness.

Line 212: Add « the reader » after « refer »

Line 220. LP can't measure negative plasma densities!!!. The processing provides negative densities which obviously points to the limitations of the used algorithm.

Line 221: Are measurements invalid or is the processing invalid?

Line 230 and following
Figure 8. Indeed, the correlation for the night side is low. Looking closely at the figure, I have explored in my own way the figure. There is a pretty good correlation on the night side for one of the lobes of the scatter point distribution.  See figure below, where the red lines figure the 1:1 correlation.

On the day side, the correlation seems to be better for lower densities than it is for higher ones. In order to put forward a possible explanation, it would be required to know the nature of the surface coating of the FP, and be reminded of which of the LP is used (Au-coated or TiN-coated) for the figure.

The assumption that the plasma composition (O+ only) is relevant because the Ion density is determined rather that than Ne. It would certainly be useful to show as well the determined Ne (which is independent of the ion composition)

[Figure]

Line 239-240. The statement: « the comparison between the LP and FP ... » is not well supported by the Fig 8 results on the night side.

Line 247-248. Indeed there a few anomalies which would need to be further worked in order to validate the plasma parameters. A clear statement on the validation and the validity of the determined plasma parameters would need to appear in the conclusions (and on the data server). I am looking forward to the description of the LP calibration measurements in baseline 06.

Line 255-256. It would be needed to ascertain that the 20 000 K Te values are not a result of processor being out of limit (as is probably the case for the negative densities). A correlation between the derived spacecraft potential and the setting bias value of the applied LP bias waveform would certainly be informative.

Line 267; It would be informative for the reader to provide the range of the specific solar illumination angle. For information I am aware of a paper (currently under revision, I cannot say more) that discusses LP « measurement peculiarities » at specific solar illumination angles at both the day-night and night-day transition.

Providing the solar illumination range when the anomalies occur would be useful to the reader.

Line 282. Earlier it is said that the LP calibration would be introduced in the baseline 06. What LP calibration are you referring to here in this paper?

Line 290.
It's hard to assess the improvement made in baseline 05. A table comparing the baselines would help.

Line 294-295:
« plasma density measurements ae more accurate during higher solar activity » I can't remember a discussion earlier in the paper that allows this conclusion. Please expand.

---

## Author Response (AR1)

**Referee 1**

**General comments**

This paper presents an interesting discussion about the Swarm Langmuir Probe data quality monitoring efforts. The paper offers a few strong points that make it worthwhile to report to the community

- It proposes two modifications from the 04 to 05 baseline. These modifications, in itself, do not represent much, yet they improve data quality and availability. As I explain further down in my review, I think the major modification would be much stronger if it would be corroborated by a physical understanding of the issue rather than only be an empirical improvement.

- A comparison with the TII front plate data suggests that the night side data products are actually not so good. While I personally think this is the most relevant finding reported in this paper, I have the impression that the authors do not sufficiently emphasize its importance.

The scientific basis of the paper is solid, although I want to raise a few questions (see my specific comments below).

The overall structure of the manuscript is logical, with a few exceptions as indicated below. I recommend the authors to introduce clear definitions of "data quality", "data availability", etc.. Some of these concepts are mixed up in the text and that does not help the clarity of the presentation. The paper suffers from many language and typographical errors; I list some of them at the end of this review without any attempt at being complete.

In conclusion, I do have a substantial number of comments and questions. Addressing these issues would likely make the manuscript suited for publication.

**1.Answer:** We thank the referee for his careful review, and for the useful and stimulating comments. We understood that some part of the text could be actually improved and we tried to follow his suggestions as discussed in the following. All the changes introduced in the manuscript are reported in bold and referenced in the answers.

**Specific comments**

In the introduction the authors list some of the most noteworthy of Swarm's accomplishments up to now. This does not really prepare the reader for the contribution that is presented in the manuscript, so I would consider this as largely superfluous material; the reference to the Swarm publication list would be sufficient – unless the authors would indicate, for the cited contributions, what role the Langmuir probes have played in these accomplishments. That would then raise the reader's awareness that these data have been proven to be of considerable scientific importance and thus that quality control is an absolute necessity.

**2.Answer:** We thank the referee for this useful comment. We have modified the introduction part emphasising the role of LP measurements in the different cited studies. Also, we added a sentence to highlight the importance of the quality control activities to maintain a high instrument performance and data quality (lines 32-66 and 74-81).

I was puzzled by the explanation (around line 80) of the difference between low and high gain, even after consultation of Knudsen et al. (2017) – in any case I suggest to add that reference here explicitly. Later on, at the beginning of section 3.1.1, the authors give a more precise

explanation. I strongly recommend to move that explanation here; this will considerably improve the readability of the text.

**3.Answer:** Following this suggestion we have moved the explanation in section 3.1.1 in this part of section 2, adding also the reference to Knudsen et al 2017 (lines 101-102).

The end of the paragraph on line 81 is a surprising one. The authors say that the use of TiN was questioned and that therefore one of the probes received an Au coating. But then they end the paragraph with a statement about the uncertainty regarding the preservation of the Au coating. I would expect the final statement to address whether the TiN coating is still preserved, because that was the initial concern.

**4.Answer:** Following this suggestion we have added a sentence that both probes don't seem to suffer from any degradation because of oxidation. Both methods, nitration and gold-plating seem to work as far as we can tell (lines 113-115).

I am curious to know how geomagnetic indices are used to process the LP data (line 94). Could the authors expand on this?

**5.Answer:** We are grateful to the referee for this comment because we realised that there was an error in the text. Actually, geomagnetic indices are not used in LP data processing. The sentence has been modified to reflect the actual input of auxiliary data in the LP PLASMA processor (line 120).

On line 100, the authors state that interpolating the 2 Hz data product at exact UTC leads to the 1 Hz sampled data. Is linear interpolation used? To what extent is linear interpolation justified as the signal likely contains faster time variations? I would expect that one would first perform a smoothing or filtering in order to remove time variability faster than 1 s before doing such an interpolation. Does one interpolate the densities and temperatures or their logarithms – which is more appropriate for quantities that vary over orders of magnitudes and that are always strictly positive? One could also interpolate the measured currents, and then do the processing with those interpolated values ...

**6.Answer :** In the ripple (or, equivalently, harmonic) mode the measurements for each plasma data "point" occurs over a period of about 400 ms. The ion density estimate is from only one of the three current/admittance measurements which still stretch over a period of about 100 ms. By correlation with the 50 Hz magnetic data the sample "points" of the ion density has been determined to be at 167 and 696 ms into the full second. Not surprisingly the correlation peak is quite broad. The interpolation of Ni and Te to the full second n is just linear between the n-1.696 and n.167 points.

The referee's suggestions are very reasonable and we might consider them as an improvement in a future update of the data processing. But we fear that any positive effects would be swamped with the uncertainties in the sampling timing and other data noise. The purpose of the interpolated data set is mainly convenience for users who need to correlate with the 1 s magnetic data.

We have replaced "exact UTC" by "full UTC second" and added "simple linear interpolation" to emphasize better that the method is quite approximating (lines 126-127).

Section 3.1.1 gives a clear explanation of the old algorithm and of the new one. It presents an evaluation of the changes and indicates that this is an improvement because of the better

correspondence with ground-based measurements. However, I miss a fundamental point: Why do the low and high gain measurements differ? As the authors have indicated, the difference is a resistor in the measurement circuit. Is there a physical underpinning of the measured difference? I believe significant effort should be invested into this. After all, this change in electron temperature calculation is the main point of improvement in data quality upon which the whole paper is built, so it has to be well documented and justified.

**7.Answer:** We agree with the referee's comment. The two probes of each Swarm satellite are different with respect to

- the probe material: the left probe is made of TiN, the right probe of gold-plated Ti;
- the position on the spacecraft with a distance between probes of only about 30 cm;
- the electronic gain, which is configurable between high and low and has been, except for test periods, always opposite between probes.

By switching the gain between the probes we determined that the gain seems to be the main cause of the differences, the others seemingly having only small effects. When the density is low, then a low gain leads to very small currents, which are difficult to measure and electronic noise and offsets become significant, especially for the ripple mode. The high gain configuration should be much more accurate at low density, but it overflows for high density. When designing the instrument the expectation was that the results from high and low gain configurations would approximately converge for a range of intermediate density. In spite of still ongoing investigations we do not have yet a clear explanation why this is normally not observed.

We prefer to not expand the discussion on this topic to avoid describing investigations which may not bring a real conclusion. We added a sentence to describe the status of the related investigations (lines 156-159).

I would welcome a clear definition of "data quality" as it plays such a central role in this paper. My intuitive understanding seems to be at odds with that of the authors. For instance, I read on line 213: "A larger percentage of invalid measurements obviously indicates a poorer data quality." That is not evident to me. One could argue otherwise: "if one obtains a higher percentage of invalid measurements, one is apparently able to catch very well those situations where the measurement process fails, so that one can have more confidence in the remaining data." Indeed, if there is an ADC overflow, that measurement clearly is not reliable, but that does not immediately say anything about the quality of the measurements performed before or after.

**8.Answer:** The data quality is the goodness of the data product as output of a processing process. The data quality can be qualified by comparison with other dataset (in-situ or ground measurement), validation with numerical or empirical models, or derived by statistical data analysis of the product itself. In our definition of data quality, if the measurement is subject to a low level of errors or data contamination derived from statistical analysis or known issues, or/and has a high agreement with other dataset (model or spacecraft observations) then the quality of the data is considered good. We have added this definition in the manuscript (lines 79-82).

The authors discuss the evolution of quality with the solar cycle. How certain are they that an apparent systematic trend with the cycle does not mask detector aging (such as cumulative damage to LP coating, consequences of nanodust impacts, etc.)? Wouldn't one need at least a complete solar cycle to evaluate this? The topic of detector aging is only briefly touched

upon. I think it deserves more attention, as this would be one of the major aspects of instrument quality monitoring. Such a discussion could be part of a more extended discussion section (which is rather short at present).

**9.Answer:** We thank the referee for this useful and stimulating comment. This is indeed a very interesting analysis. We are using some parameters to monitor the detector performance and so far we do not have any evidence of a degradation with respect to the solar cycle. However this analysis certainly deserves more attention and we should expand the set of parameters that we are monitoring before coming to a clear conclusion. We reserve to further investigate this aspect and present the results in a future work,

A very interesting point is the comparison of LP and FP densities presented in Figs. 6 and 8. On line 235ff the authors say that the FP processor does not need any assumptions regarding ion composition, while the LP processor does. But then line 236 states that measurement differences between both are due to different assumptions – that contradicts the statement that FP uses no assumptions at all. This is an important point, because it suggests that the night side FP density measurements are considerably more reliable than the LP densities there. And consequently, in view of the observed relatively poor correlation, I do not understand the assessment on p. 245 "The Swarm LP measurements are of very good data quality" – shouldn't this be qualified somewhat, e.g. restricted to the day side? The conclusions section does list this problem as one that will drive future attempts for improvement.

**10.Answer:** We agree with the referee that the sentence was misleading, thus we adjusted the sentence specifying that the difference in the FP and LP measurements may be due to the assumption made for LP processor. We removed part of the sentence at the beginning of section 4, because the details of the goodness of data are summarised in the conclusion part.

**Detailed issues**

Abstract: The abstract reads well, explaining that the paper discusses the quality control approach. I suggest to add a sentence that states how good the data quality is, before then saying that there is an anomaly. Please also do not use the LP abbreviation in the abstract as it is not explained before.

**11.Answer:** We agree with the comments reported by the referee and we adjusted the abstract accordingly.

Throughout the paper there are numerous issues with punctuation – lots of unnecessary or misplaced commas. I have indicated only some of them below.

Line 18: divers -> diverse ok

Line 21: satellite' s -> satellite's (remove blank space) ok

Line 22: plasma and electric field -> electric field and plasma (so that the order of both corresponds to the names of the sensors mentioned before)

Line 33: reveled -> revealed ok

Line 41 mission, -> mission (drop comma) ok

Line 45 fluctuations, -> fluctuation (drop comma) ok

Line 45: described -> describes ok

Line 50: largely -> extensively (?) ok

Line 54: remind -> refer ok

Line 70: instruments -> instrument's or instruments' ok

Line 91: to simple flow-chart -> to the simple flow-chart ok

Line 95: essentials -> essential ok

Line 98: remove comma (or have it after "Finally") ok

Line 106: expectantly: not sure what is meant by this; I propose to drop this word ok

Line 108: "This data is" or "These data are" ok

Line 112: In the next two sessions are described -> The next two sessions describe ok

Line 116: I do not think that explaining the file naming convention that is used by the team is of any value to the reader; I propose to drop this sentence. ok

Line 121: baseline lower than 04 are -> "baseline lower than 04 is" or "baselines lower than 04 are" ok

Line 124: ESA (2018, 2020b) -> (ESA, 2018, 2020b) ok

Line 126: consisting on -> consisting of ok

Line 126: the first item you mention is not a "difference". Replace by "consisting of an updated electron temperature computation" or something of that style. ok

Line 132: differ which –> differ, which ok

Line 137: appropriated -> appropriate ok

Line 145: drawback, -> drawback ok

Line 147: region -> regions ok

Line 176: where -> was ok

Line 178: PLASMA -> the PLASMA ok

Line 179: occurred -> that occurred ok

Line 184: impact in -> impact on ok

Line 186, 193: remove comma ok

Line 195: sun spot -> sunspot ok

Line 197: in-orbit -> in orbit ok

Line 224, 226: remove comma ok

Line 268: In the next baseline will be introduced -> The next baseline will introduce ok

Line 282: the calibrating the LP measurements yield -> the calibration of the LP measurements yields ok

Caption of Fig 3: each one-degree bins in latitude -> each one-degree bin in latitude ok

Figure 5: I see little value of showing the entire sunspot record. The figure would be much more clear if all panels would focus on the Swarm mission period. We revised Figure 1

**Referee 2**

Reviewer report

This paper addresses the validation of the Swarm Langmuir Probe (LP) measurements and of the derived plasma parameters.

This is a very interesting and useful paper that addresses a topic that was never addressed, to my knowledge, in such depth. The authors must be commended for the efforts made to improve and characterize the Swarm LP measurements and the derived parameters.

The Swarm LP data are already being extensively used by the ionospheric community, and it is clear that, when the derived plasma parameters will be fully validated and properly flagged when uncertain, the swarm data will be an excellent reference for Langmuir probe data validation. It would also facilitate further novel studies of the ionospheric processes.

My main concern in the validation approach is that the peculiar position of the LPs, close to the spacecraft skin (within 10 cm) does not address the limitation of the quality of the measurements, hence the validity of the derived parameters, due to the fact that the LPs are most likely immersed in the spacecraft plasma sheath. The same is true also for the interpretation of the FP measurements, although not a direct subject of this paper.

**1.Answer:** We thank the reviewer for the accurate review and for expressing the interest on the subject reported in this paper. We found very stimulating and interesting comments on this report that we tried to address as reported in the following.

The reviewer's concern for the LP positions relatively close to the spacecraft, possibly within its plasma sheath, may very well affect the Te estimates. But for Ne estimates we had early on decided to rather use the ion current, obtained at a negative bias repelling the electrons. In this case, owing to the high orbital velocity of about 7.5 km/s, the existence of a plasma sheath around the spacecraft is less important. The current is simply determined by ions that ballistically hit the probes. O+ ions have an energy of about 4.8 eV in the spacecraft frame, and typically overcome the sheath potential. The data show that the ion current (and admittance) is relatively free from disturbances and spikes compared to the electron current at positive bias. Also the faceplate is kept at negative bias, Te or the spacecraft potential are not measured with it. Therefore the 16 Hz current is also "good" in spite of the plasma sheath. The downside of this approach is that we need to assume a fixed ion composition (O+), and have to ignore possible along track plasma drifts that would modify the effective spacecraft velocity. Such ion drifts would have to be relatively large to significantly influence the density estimate, but they actually may occur at high latitudes. Indeed a comparison with 16 Hz data, the "true" electron density, and other external density estimates show offsets that vary systematically. This could be related to the caveats above, and is under investigation (as projects within the SWARM DISC).

In the revised version of the manuscript we improved the description of the method and possible issues with it (see Sections 2 and 3).

All the changes introduced in the manuscript are reported in bold and referenced in the answers.

Regarding the title, I would suggest the following modifications:

Swarm Langmuir Probes' data quality validation and plans for future improvements

**2.Answer:** We are grateful for this suggestion that we kindly accept. The title has been modified accordingly.

Detailed comments, including editorial suggestions, are written in the attached document.

 Reviewer's detailed comments

§ Line 2: « very accurate » needs to be quantified. One knows that LP measurements are not that accurate in absolute terms.
**3.Answer:** We agree with the referee's comment and we replaced "very accurate" with "in situ" in line 2.

Line 4: Clarify the « operational use » that is being made of LP measurements

**4.Answer:** In that sentence operational was intended more as technical applications. An example of the technical application would be the development of some ionospheric models, or the usage of LP measurements as input for more complex data processing algorithms to derive other related quantities, as for example the ionospheric plasma irregularity index derived in L2 Swarm data products (IPIR). However we realised that the term "operation" is misleading and we replaced it with "technical" in line 4.

Line 9: Replace « conduced » by « conducted »
**5.Answer:**We agree with this correction and we modified the word accordingly.

Line 9: Note sure that how the feed back from the user community is addressed in the manuscript. It should be noted that the intend to involve the feedback from the science community is excellent and will certainly contribute to the data product quality.
**6. Answer:** We completely agree with the referee that the users community feedback are essentials to improve the Swarm data product quality. Most of this feedback actually results in recommendations discussed at yearly Swarm Data Quality workshops which are used as a roadmap for future data processing algorithm evolution. To emphasise the importance of the feedback from the community we have added a sentence in line 74 to 81.

Line 12: add « in the derived plasma parameters » after « improvements »
**7. Answer:** We agree with this suggestion and the sentence has been modified accordingly.

Line 17: I am wondering whether the accurate determination of the plasma parameters contributes to the improvement of the magnetic field measurement analysis? If not, please ignore my question, if yes it would be interesting to say a few words on the subject.
**8. Answer:** Currently, LP measurements are not used to improve magnetic field analysis. It had been suggested that a spacecraft plasma wake might affect the magnetic field measurements. But an early study showed that the differences between data from both magnetometer types on each Swarm satellite (VFM and ASM) are not strongly correlated with the density (they seem to be rather correlated with temperature variations caused by the sun shining on the booms)

Line 18: Replace « divers » by « diverse »
**9. Answer:** We thank the referee for this correction which is now implemented in the manuscript.

Line 31: Rephrase « insights on the space weather »
**10. Answer:** We modified this part by using "brought more understanding". We thank the referee for that comment.

Line 33: Replace « reveled » by « revealed »
**11. Answer:** We actually removed this sentence from the new version of the manuscript because we realised that we were referring to a paper where EFI-TII data only were used.

Line 41: remove the comma after « mission »
**12. Answer:** Removed, we thank the referee for this correction.

Line 41: I doubt the word « climatology » is the right word. Would « meteorology » be more appropriate? as it seems related to space weather effects.

**13. Answer:** We thank the referee for this comment that actually brought to us the following consideration. The word "climatology" identifies the analysis of changes in ionospheric phenomena, while "meteorology" is generally intended as the study of these phenomena on a daily basis, in order to register the changes of the ionospheric parameters. This is a general definition which, even if it can be a little confusing, is generally accepted by the community. For this reason we preferred to keep the word "climatology" in that sentence.

Line 42: add the word « performed » after « variations »

**14. Answer:** In agreement with the concept suggested by the referee we added the word "observed by" in that sentence.

Line 45: replace « this new statistics » by « This new statistical study »

**15. Answer:** We agree with this suggestion and we modified the sentence accordingly.

Line 47: Precise « Antarctic » which usually refers to the continent

**16. Answer:** We agree with this observation and we specified the region by using "southern polar cap".

Line 59: Provide a reference to the « PLASMA operational processor »

**17. Answer:** We thank the referee for this observation. We actually use the reference for the PLASMA processor algorithm in the sentence just after the one here mentioned.

Line 60: Is « PLASMA processor » the same thing as « PLASMA operational processor »?

**18. Answer:** We are grateful for this question that gives us the chance to specify the difference between the two. In line 60 we refer to the PLASMA processor algorithm, which is identified as the scientific code developed to process the data. The PLASMA operational processor basically consists of the implementation of the PLASMA processor algorithm in a dedicated server able to maintain a constant data flow and expected processing performance.

Line 61: add a comma after « research »

**19. Answer:** We agree with the suggestion that has been implemented accordingly.

Line 63: Precise if you refer to « data quality » or « Plasma parameters quality »

**20. Answer:** In that sentence we were actually referring to data quality, not strictly related to plasma parameters. As discussed in section 3.1 (and subsections), an improvement in the data quality is for example the better data flagging method applied in the new baseline.

Line 68: Explain « admittance »

**21. Answer:** Admittance is a measure of how easily a circuit or device will allow a current to flow, which is the reciprocal of impedance. We have added this definition in the manuscript (line 87-88).

Line 69: Precise « given biases »

**22. Answer:** We thank the referee for this comment. Here, given biases are intended as predefined values of the voltage. However these values can be changed according to the operational requests or triggered to optimize the plasma parameter derivation. In order to not add

confusing information and to not elengute this description much, we prefer to avoid further definitions and  we simply used the reference of Knudsen et al 2017 for further details.

Line 71-73: The position of the LP sensor real close to the Satellite skin probably affects the accuracy of the spacecraft potential measurements (and probably to a certain extent plasma density and temperature) , as the LPs are most likely inside the plasma sheath of the satellite. The Debye Length of the effect should be discussed. This is a potential issue which is not addressed in the paper. Perhaps it was addressed in a reference publication. Its impact on the accuracy of the plasma parameters measurements should be addressed in this paper.

**23. Answer:** We are grateful for this comment. Indeed, within a typical orbit the Debye length exceeds the probe-spacecraft distance at low densities, especially at high latitudes near the dark pole, in the mid-latitude trough/sub-auroral polarization streams and sometimes inside equatorial plasma bubbles. As already replied above, we think that the density estimates based on the ion current are not affected by the plasma sheath of the satellite. The estimates of Te and the spacecraft potential Vs could be affected, and this is mentioned in the revised manuscript., citing relevant publications, e.g. Wang et al., (2015) (see lines 94-96).

Line 74 (and 77): Replace « signal » by « current »?

**24. Answer:** We thank the referee for this observation. Here the signal is more intended as the response of the instrument during  in situ measurements, while current can be considered as a derived quantity in a certain way. Thus we think it is more appropriate to maintain the word "signal"instead of replacing it with "current".

Line 78: replace « interfere » by « interfer »

**25. Answer:** Here we have to disagree with the referee, because interfere is more frequently used to intend some interference.

Line 82: What was the rationale for the gain switch between the two probes in 2019?

**26. Answer:** The rationale for probe switching was mainly to compare both probes. If their gains are equal, then their surface material is still different (TiN vs Au), and their position at the spacecraft. The effects of both might become more obvious if the gains are equal. The test has been done for a limited time in the solar minimum, when ADC overflows are relatively fewer. The comparison of the data has not yet been concluded.

Line 84-86: The sentence « Out of concern »… ; please discuss whether if there is a difference in terms of degradation between Au and Ti due to Atomic oxygen. Knowing the subject, I may add that, one of the concerns, was also the oxidation of TiN on ground.

**27. Answer:** We thank the reviewer for this comment. A laboratory test of an LP flight spare had shown that indeed the TiN surface contained some TiO, i.e. oxidation had occurred in the normal atmosphere. The PI of the LP on the DEMETER satellite, J.-P. Lebreton had expressed his suspicion that the DEMETER data were possibly affected by oxidation of Ti in orbit (oral communication).  We decided to replace one of the probes, but because of already completed

vibration tests we could not use a heavier material such as brass which is often used for gold-plating. Thus the only option was to gold-plate the Ti. We added a sentence in lines 111-112.

Line 87: Explain what « testing before launch » was made
**28. Answer:** We thank the referee for this comment. Well, we tried to verify that the Au would stick to the probe by baking the probe in an oven to more than 300 C and putting it in an ultrasonic cleaner. There was no possibility to test in an atomic oxygen atmosphere.

Line 90 and following It's rather hard for the reviewer (and will most likely be hard for the readers) to follow the meaning of all acronyms used. A list of acronyms would be useful.
**29. Answer:** We are grateful for this suggestion. However the manuscript guidelines recommended defining the acronyms in line through the manuscript.

Line 103-104. I am myself not aware of a previous use of the HM method in space. I am wondering how the set point of the applied bias signal (and its amplitude) is selected and adjusted as satellite potential must significantly vary along the orbit. I note that it is written that « Sweep mode data are not used in the PLASMA processor… ». Has a comparison of the plasma parameters regularly obtainable from the I-V sweeps and the HM method been made.?
**30. Answer:** The bias for the ion current and admittance is fixed (-2.5 V has so far always been used). The measurement cycle for the electron currents/admittances starts with iteratively finding the bias for a very small current, which is called "zero-tracking". The iteration starts from the 0-tracked bias of the previous measurement, and normally converges within a few iterations. To this 0-tracked bias a constant offset (0.7 V) is added to determine the bias in the region of the linear (or saturated) electron current. Current and admittance, i.e. slope of the I-V curve, at this linear bias is determined. Extrapolating with help of the slope the bias for the retarded electron region, the "knee" of the I-V curve, can be reliably obtained. All this is controlled by the LP electronics on-board. At ground we then have bias, current, and admittance in the three regions of ion, retarded and linear electron regions. From these Ne, Te, and Vs are estimated. The document https://earth.esa.int/eogateway/documents/20142/37627/swarm-level-1b-plasma-processor-algorithm.pdf/bae64759-b901-d961-4d18-0a5b317f8c12 describes the algorithm in more detail.

We had hoped to merge sweep mode data with the HM results, but they differ causing "spikes" in the time series of Ne, Te, and Vs every 128 s when a sweep derived data point occurs. A comparison of sweep and HM mode data has not yet been published. The preferred strategy is to calibrate the HM data using external data sets as in Lomidze et al. (2018), and Smirnov et al. (2021).

Line 112: Replace « sessions » by « sections »
**31. Answer:** The suggestion has been implemented in the manuscript.

Line 115 and following Would it be useful to summarize in a table, the differences between the baselines 04 and 05?
**32. Answer:** We thank the referee for this suggestion. We added Table 2 to summarise the main differences between the two baseline.

Line 129-130. Clarify « second shunt resistor ». What's the role of the first one?

**33. Answer:** We removed that sentence from the manuscript because it did not add any useful information.

Line 132: The sentence « In practice the values often differ which we suspect is because of the different probe gain » is problematic. The sudden jumps mentioned in line 135 should be further discussed. Also, statement in line 138-139 casts doubts on the accuracy of the measurements at the gain transition. It seems to point to the fact that the measurements performance in high and low gain are not fully understood, thus question the validation of the derived plasma parameters. It may be useful to look at this issue by looking at the regular sweeping I-V characteristics.

**34. Answer:** We thank the referee for this comment. We expanded the discussion on this point in the manuscript to explain that unfortunately, from our analysis, we do not have a conclusion yet that can support the observed difference between the two values.

Line 139 and following: Provide details on the method used to derive the ion density rather than the electron density. This should be supported by modelling the ion sheath in the vicinity of the LPs. Assumption the Ni=Ne at the LP location may be questioned.

**35. Answer:** We agree with the referee that the assumption of Ni=Ne may be not fully correct. Indeed, we are introducing on our algorithm the computation of both the variables Ni and Ne so that the users can have the possibility to compare the two and to use the two separately. This part of the algorithm is currently under development and we reserve to discuss the details in future works.

Line 147-148: « .. the region (singular) . ; are (plural) « . Correction needed.

**36. Answer:** Line modified according to the referee's suggestion.

Line 147: « Larger » than what? use « large «?

**37. Answer:** Line modified according to the referee's suggestion.

Line 176: Replace « where » by « were »

**38. Answer:** Line modified according to the referee's suggestion.

Line 178; add «, the » after « 05 »

**39. Answer:** Line modified according to the referee's suggestion.

Line 179: Add « the » after « gaps »

**40. Answer:** Line modified according to the referee's suggestion.

Line 186: replace « today » by a specific date

**41. Answer:** We replaced "today" with "onwards". Giving a specific date will reduce the data coverage with baseline 05, which will actually exist until a new baseline will be introduced (line 119-120).

Line 210: How « good » is good?. Quantify goodness.

**42. Answer:** We thank the referee for this question. However we preferred to remove the word "good" because it is always difficult to unequivocally quantify goodness. For clarity, we have added a description of the concept of data quality applied to this paper (lines 79-82).

Line 212: Add « the reader » after « refer »

**43. Answer:** Line modified according to the referee's suggestion.

Line 220. LP can't measure negative plasma densities!!!. The processing provides negative densities which obviously points to the limitations of the used algorithm.

**44. Answer:** We thank the referee for this comment. We modified the sentence to specify that the density derived from the data processing was negative (line 250).

Line 221: Are measurements invalid or is the processing invalid?

**45. Answer:** The processing algorithm returns invalid density data. We modified the sentence to reflect this aspect (line 251).

Line 230 and following Figure 8. Indeed, the correlation for the night side is low. Looking closely at the figure, I have explored in my own way the figure. There is a pretty good correlation on the night side for one of the lobes of the scatter point distribution. See figure below, where the red lines figure the 1:1 correlation.

On the day side, the correlation seems to be better for lower densities than it is for higher ones. In order to put forward a possible explanation, it would be required to know the nature of the surface coating of the FP, and be reminded of which of the LP is used (Aucoated or TiN-coated) for the figure.

The assumption that the plasma composition (O+ only) is relevant because the Ion density is determined rather that than Ne. It would certainly be useful to show as well the determined Ne (which is independent of the ion composition)

**46. Answer:** We thank the referee for this useful comment. For producing this figure we mainly used the density derived mainly from TiN probe. However, the densities derived from Tin and Au probe are pretty much in agreement (not shown). Also the faceplate surface is made of Ti. Unfortunately, the algorithm to derive the actual Ne from LP data is under development. We reserve to present in a future work a similar analysis, as soon as the Ne data will be available.

Line 239-240. The statement: « the comparison between the LP and FP … » is not well supported by the Fig 8 results on the night side.

**47. Answer:** We added a few sentences in this section to clarify our findings (lines 265-271).

Line 247-248. Indeed there a few anomalies which would need to be further worked in order to validate the plasma parameters. A clear statement on the validation and the validity of the determined plasma parameters would need to appear in the conclusions (and on the data server). I am looking forward to the description of the LP calibration measurements in baseline 06.

**48. Answer:** We thank the referee for this comment. We are indeed putting much effort to further improve the LP measurements.

Line 255-256. It would be needed to ascertain that the 20 000 K Te values are not a result of processor being out of limit (as is probably the case for the negative densities). A correlation between the derived spacecraft potential and the setting bias value of the applied LP bias waveform would certainly be informative.

**49. Answer:** We thank the referee for this suggestion. We are not reporting much details on this investigation because we do not have a clear conclusion yet, but we will certainly consider this suggestion for the ongoing investigations.

Line 267; It would be informative for the reader to provide the range of the specific solar illumination angle. For information I am aware of a paper (currently under revision, I cannot say more) that discusses LP « measurement peculiarities » at specific solar illumination angles at both the day-night and night-day transition. Providing the solar illumination range when the anomalies occur would be useful to the reader.

**50. Answer:** We thank the referee for this suggestion. However, as explained in the manuscript and in the previous point, we are not really sure about the source of this anomaly. For this reason, we think that reporting details that are not yet verified could be confusing to the reader. We reserve to discuss this topic more in future work as soon as we will reach a consolidated conclusion.

Line 282. Earlier it is said that the LP calibration would be introduced in the baseline 06. What LP calibration are you referring to here in this paper?

**51. Answer:** We have modified this sentence specifying the calibration we were referring to (line 324).

Line 290. It's hard to assess the improvement made in baseline 05. A table comparing the baselines would help.

**52. Answer:** We agree with the referee and we added Table 2 to summarise the main updates introduced in baseline 05.

Line 294-295: « plasma density measurements are more accurate during higher solar activity » I can't remember a discussion earlier in the paper that allows this conclusion. Please expand.

**53. Answer:** In order to not repeat the discussion, we have added the relevant Figure as reference (line 328).